# How Molecular and Ancillary Tests Can Help in Challenging Cytopathology Cases: Insights from the International Molecular Cytopathology Meeting

Elena Vigliar [1,†], Claudio Bellevicine [1,†], Gennaro Acanfora [1], Allan Argueta Morales [2], Anna Maria Carillo [1], Domenico Cozzolino [1], Mariantonia Nacchio [1], Caterina De Luca [1], Pasquale Pisapia [1], Maria D. Lozano [2,‡], Sinchita Roy-Chowdhuri [3,‡] and Giancarlo Troncone [1,*,‡]

1   Department of Public Health, University of Naples, "Federico II", 80131 Naples, Italy;
    elena.vigliar@unina.it (E.V.); claudio.bellevicine@unina.it (C.B.); gennaro.acanfora95@gmail.com (G.A.);
    a.m.carillo@virgilio.it (A.M.C.); cozzolino.domen1994@libero.it (D.C.); mariantonia.nacchio@unina.it (M.N.);
    caterina.deluca@unina.it (C.D.L.); pasquale.pisapia@unina.it (P.P.)
2   Department of Pathology, Clínica Universidad de Navarra, University of Navarra, 31008 Pamplona, Spain;
    aargueta@unav.es (A.A.M.); mdlozano@unav.es (M.D.L.)
3   Department of Pathology, The University of Texas MD Anderson Cancer Center, Houston, TX 77030, USA;
    sroy2@mdanderson.org
*   Correspondence: giancarlo.troncone@unina.it
†   These authors contributed equally to this work.
‡   These authors contributed equally to this work.

**Abstract:** Over the past decade, molecular cytopathology has emerged as a relevant area of modern pathology. Notably, in patients with advanced-stage cancer, cytological samples could be the only material available for diagnosis and molecular biomarker testing to identify patients suitable for targeted therapies. As a result, the contemporary cytopathologist's role extends beyond morphological assessments to include critical skills such as evaluating the adequacy of the cytological samples and managing these specimens for molecular testing. This case collection can be a valuable source of insight, especially for young pathologists, who should learn to combine the opportunities offered by molecular biology with the basis of morphological evaluation.

**Keywords:** cytopathology; molecular cytopathology; FNA; ancillary test

## 1. Introduction

Over the past decade, molecular cytopathology has emerged as a relevant area of modern pathology; indeed, the number of molecular tests conducted on cytological material has substantially increased, thanks to the ease of preparing high-quality nucleic acids and the remarkable versatility of the different cytological preparations.

Notably, in patients with advanced-stage cancer, cytological samples could be the only material available for diagnosis and molecular biomarker testing to identify patients suitable for targeted therapies. Beyond its role in predicting responses to targeted cancer treatments, molecular profiling of cytological samples also plays a role in diagnostics, assisting in categorizing cytology classes with undetermined findings into low and high malignancy risk categories.

As a result, the contemporary cytopathologist's role extends beyond morphological assessments to include critical skills such as evaluating the adequacy of cytological samples and managing these specimens for molecular testing.

Since 2010, the international Molecular Cytopathology Meeting provides a continuous educational effort to update cytopathologists on novel molecular targets and testing methodologies and to unravel the challenges of molecular testing on small tissue samples.

During the 12th annual meeting (11–12 December 2023, Naples, Italy) directed by Professor Giancarlo Troncone and co-chaired by Professor Sinchita Roy-Chowdhuri and Maria D. Lozano, a digital slide session on cases in which molecular tests were pivotal for the final cytological diagnosis and patient management was held by junior fellows presenting unknown cases and actively involving the audience

This case collection can be a valuable source of insight, especially for young pathologists, who should learn to combine the opportunities offered by molecular biology with the basis of morphological evaluation.

## 2. Case 1

As a referral center for predictive testing on cytological and histological specimens, we received a lung fine needle aspiration (FNA) sample collected in a peripheral hospital for the immunocytochemical (ICC) assessment of PD-L1. The original cytological diagnosis was high grade squamous cell carcinoma, probably primary of the lung. One Diff-Quik stained smear and one CB were submitted to our institution. The primary cytopathologist did not perform any ICC. The smear showed, in a necrotic background, epithelial cells arranged in solid groups without evidence of mucin or keratin (Figure 1A); the cells exhibited large cytoplasm and pleomorphic nuclei with irregular nuclear membranes. The same morphological features were seen on the CB hematoxylin\eosin-stained section (Figure 1B).

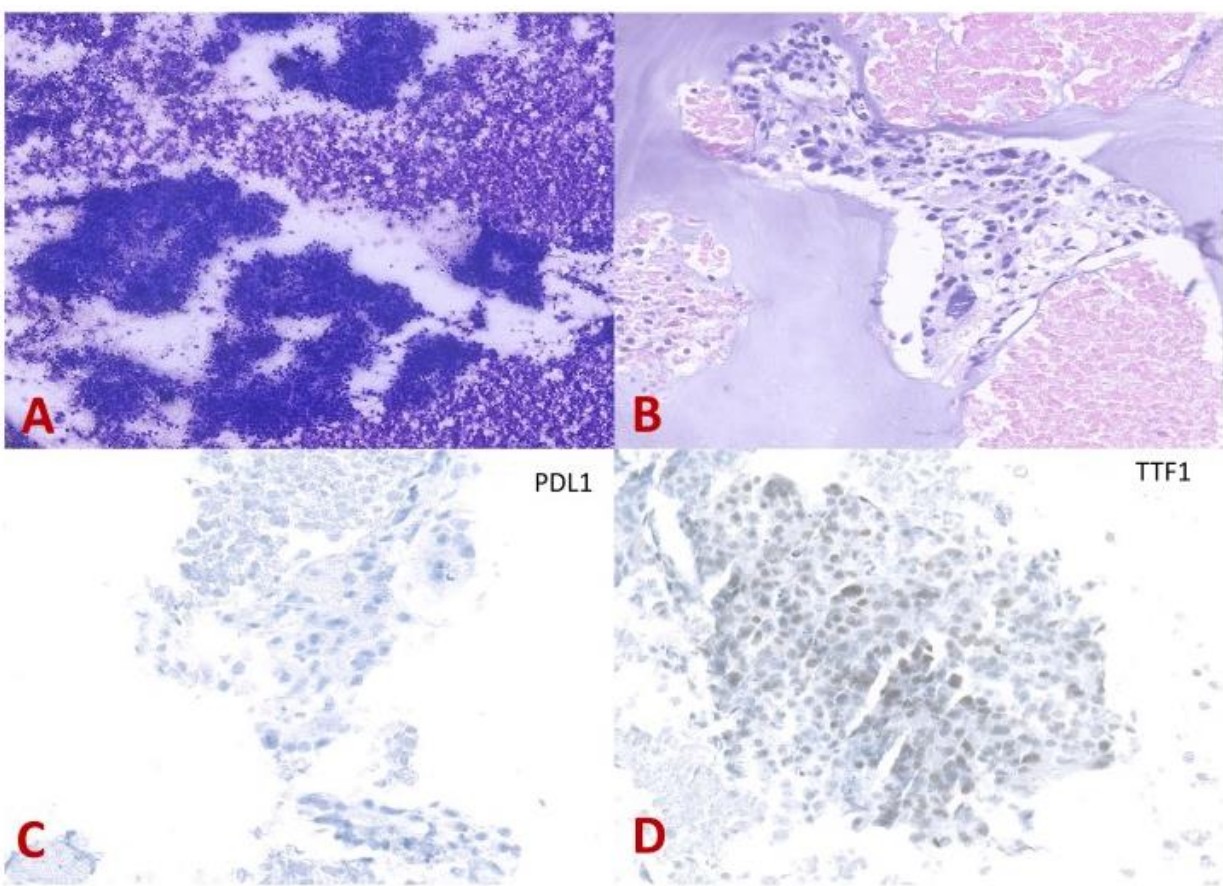

**Figure 1.** Cohesive group of atypical epithelial cells in direct smear (**A**) (Diff-Quik staining, ×10), and in cell block section (**B**) (hematoxylin/eosin staining, ×20). (**C**) No granular or linear membrane signal was observed in neoplastic cells (PDL1 immunocytochemistry, ×20). (**D**) TTF1 staining showed a weak positivity (×20) (**D**). Immunoperoxidase detection with diaminobenzidine.

The CB was deemed adequate for the assessment of PD-L1 and ICC was performed with the companion diagnostic kit SP263 assay (Ventana) according to the manufacturer's

instructions. No granular or linear membrane signal was observed, so the proportion of the positive neoplastic cells among the total viable neoplastic cells (tumor proportion score, TPS) was <1%. (Figure 1C). Then, the oncologist requested comprehensive biomarker testing [1]. Additional CB 5 μm sections were used to perform the next generation sequencing (NGS) analysis (Genexus, Thermo Fisher Scientific, Waltham, MA, USA) by using an Oncomine Precision Assay (OPA, Thermo Fisher Scientific) that is able to detect 78 variants across 50 key genes in both tissue and liquid biopsy specimens. A minimum of 10 ng of nucleic acids input is required. A point mutation in exon 21 (L858R) of the EGFR gene was detected. L858R is an uncommon mutation in squamous cell lung carcinoma patients and it is reported in 0.91% of cases [2]; therefore, a morphological re-evaluation was performed. Pseudoglandular differentiation was observed; TTF1 staining was then performed, showing weak positivity (Figure 1D). Accordingly, a diagnosis of non-small cell carcinoma of the lung, favoring adenocarcinoma, was rendered [3].

Several critical issues emerged from the present case: (1) in a case of poorly differentiated non-small cell lung cancer on a small tissue sample, either a biopsy or cytology specimen, a limited ICC panel should be performed to discriminate between squamous cell carcinoma and adenocarcinoma while saving material for biomarker testing; (2) in this scenario, PDL1 ICC assessment and comprehensive biomarker testing were requested in a sequential approach. However, these sequential management approaches have the disadvantage of being time consuming and also causing exhaustion of the sample.

Instead, a comprehensive approach to molecular testing, in which all biomarkers are simultaneously evaluated, ideally via NGS and ICC, optimizing time and the limited material, is recommended [4]. To this end, a proper classification is mandatory to give the patient the best treatment. Furthermore, recent recommendations have suggested reflex biomarker testing, in accordance with local circumstances, availability, and adherence to a protocol, defined and agreed on by the multidisciplinary team [4].

### 3. Case 2

A CT scan of a 59-year-old man, former smoker, revealed a solid 10 × 10 cm mediastinal mass that infiltrated the superior vena cava and brachycephalic trunk. He also presented with a tumor in the superior and middle lobules of the right lung (4.8 × 4.7 cm) with no apparent relationship with the mediastinal mass, as well as multiple adenopathies in the mediastinum. An endoscopic ultrasound FNA (EUS-FNA) and an endoscopic ultrasound fine needle biopsy (EUS-FNB) of the mediastinal mass and an EUS-FNA of the hilar adenopathy, with rapid on-site evaluation (ROSE) were performed.

Only smears were available from the adenopathy, which were highly cellular with a necrotic background and showed cells distributed on sheets or individually, with clumped or dark nuclei, focally pyknotic; keratin was observed. The p40 immunostaining was diffusely positive; hence, a final diagnosis of metastasis from a squamous cell carcinoma of pulmonary origin was rendered.

From the mediastinal mass, both smears and CB were available. The smears were highly cellular, with small, round, monotonous cells, some of which were spindle, with hyperchromatic nuclei and a scant cytoplasm, accompanied by a myxoid stroma (Figure 2A). The CB showed cartilaginous foci and a prominent hemangiopericytomatous growth pattern featuring an abrupt transition between the cellular component and these cartilaginous foci (Figure 2B). Necrosis and occasional mitotic figures were also evident. ICC performed on the CB demonstrated positivity for SOX9 and CD99 and negativity for S100, cytokeratins, and p40 (Figure 2C). A FISH for HEY1-NCOA2 was performed on a cytological smear. The FISH analysis was conducted using the ABNOVA HEY1/NCOA2 DY translocation FISH Probe (Catalog no. FT005) (ABNOVA, Ref 16043545-100 μL) targeting the location 8q21.13, 8q13.3. Under normal conditions without translocation, two separate green signals and two separate red signals are observed per nucleus. A translocation event is identified by the presence of one independent green signal, one independent red signal, and a red/green fusion signal in more than 30% of the analyzed nuclei. In our case, a total

of 45% of tumor cells presented fusion signals (Figure 2D), indicating the rearrangement of HEY1/NCOA2, which confirmed the diagnosis of mesenchymal chondrosarcoma. The evidence of rearrangement of HEY1–NCOA2 confirmed the diagnosis of a mesenchymal chondrosarcoma.

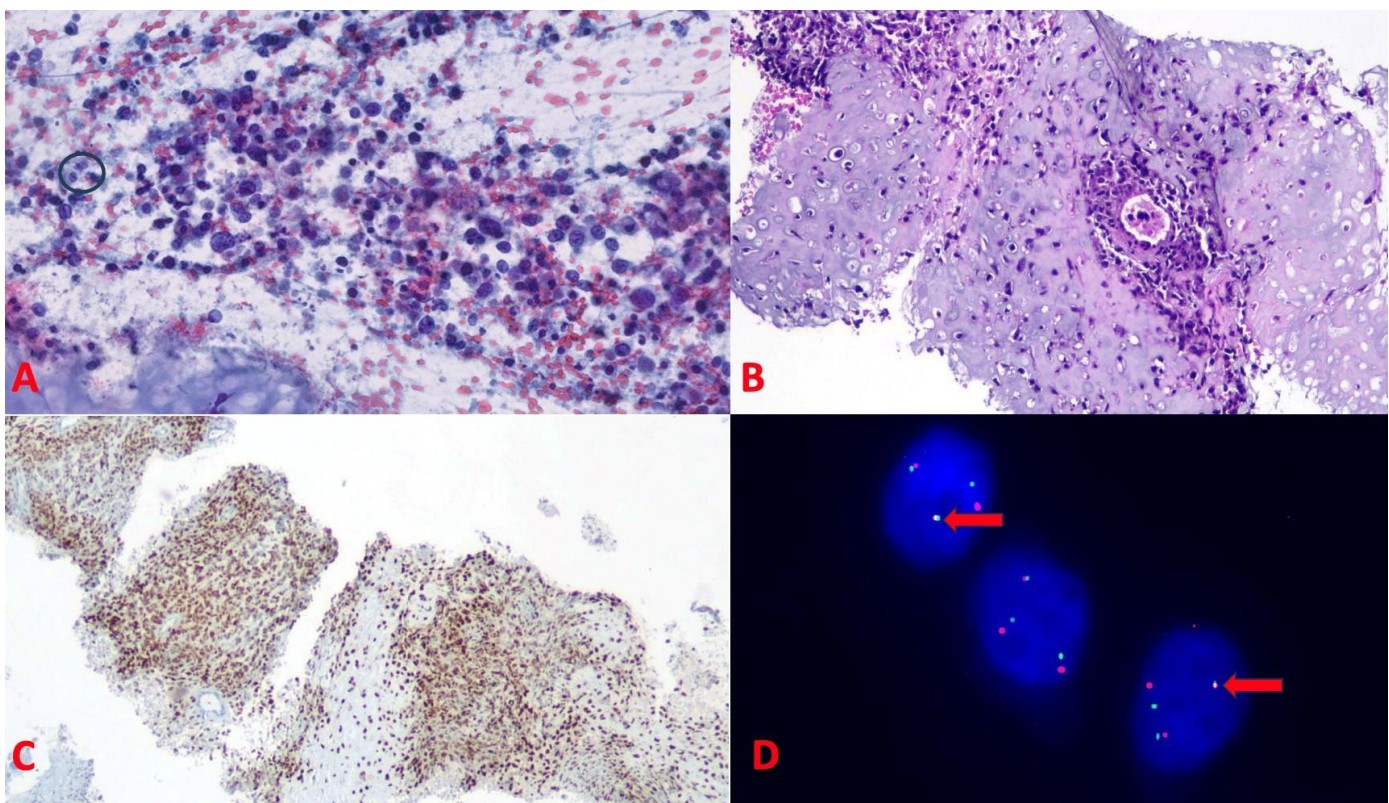

**Figure 2.** (**A**) Cellular smear with small round blue cell tumors with hyperchromasia and scant cytoplasm (20×). (**B**) Cell block exhibits small round blue cell tumors, with abrupt transition to cartilaginous foci and hemangiopericytomatous growth pattern (20×). (**C**) Immunohistochemical staining of tumor cells with SOX9 (20×). (**D**) Fluorescence in situ hybridization (FISH) analysis on cytological smear using a HEY1/NCOA2 translocation probe to detect HEY1–NCOA2 fusion. The fused signal (visible as one yellow dot (red arrow) instead of a green dot/red dot side by side for the separate probes) is seen in positive cells (100×).

As shown in this case, the addition of ROSE significantly enhanced any diagnosis approach involving FNA, and it streamlined the process of obtaining suitable material for ancillary techniques, such as ICC and molecular testing [5,6]. This approach is pivotal, especially in challenging scenarios like the present case, in which two primary tumors, with different morphological features, were diagnosed. Moreover, each preparation type should be maximized; in fact, as far as the possibility of mesenchymal chondrosarcomas, although positivity for S100, CD99, SOX9, NKX3.1, and negativity for cytokeratins can be observed [7,8] the presence of HEY1–NCOA2 rearrangement is confirmatory of the diagnosis [9]; the rearrangement can be evaluated by FISH, which due to the higher DNA quality, nuclei preservation integrity, and guarantee of a more efficient hybridization, can be successfully carried out on cytological smears [5].

## 4. Case 3

A 19-year-old man affected by a recent growth in the right parotid mass was referred for a FNA at our institution. The interventional cytopathologist performed the FNA under ultrasound guidance, staining the first smear with Diff-Quik for the ROSE to evaluate the adequacy of the sample. The smear was hypercellular and composed of solid and papillary

groups of polygonal cells with a low N:C ratio and indistinct cell borders in a background featuring scant metachromatic extracellular material (Figure 3A,B). At higher magnification, the cells showed abundant vacuolated pale cytoplasm and uniform round eccentrically placed nuclei with indistinct nucleoli. Based on the cytomorphology observed during the ROSE, the interventional cytopathologist performed an additional pass to ensure a CB preparation. Thus, immunocytochemistry on CB sections was performed, showing intense and diffuse S-100 positivity (Figure 3C). The FNA was classified as salivary gland neoplasm with uncertain malignancy (SUMP) according to the Milan System for Reporting Salivary Gland Cytopathology (MSRSGC) [10]; in the final report, a comment discussing the possibility of a secretory carcinoma was included. The patient underwent surgical resection; the histopathology showed a neoplasm with a solid and microcystic pattern (Figure 3D) with the same cytological features encountered in the FNA. The immunohistochemistry showed that the neoplasm was intensely positive for S100, cytokeratin 7, mammoglobin and GATA3, while it was negative for p63 and DOG1 immunostaining. Thus, a histopathological diagnosis of secretory carcinoma was rendered.

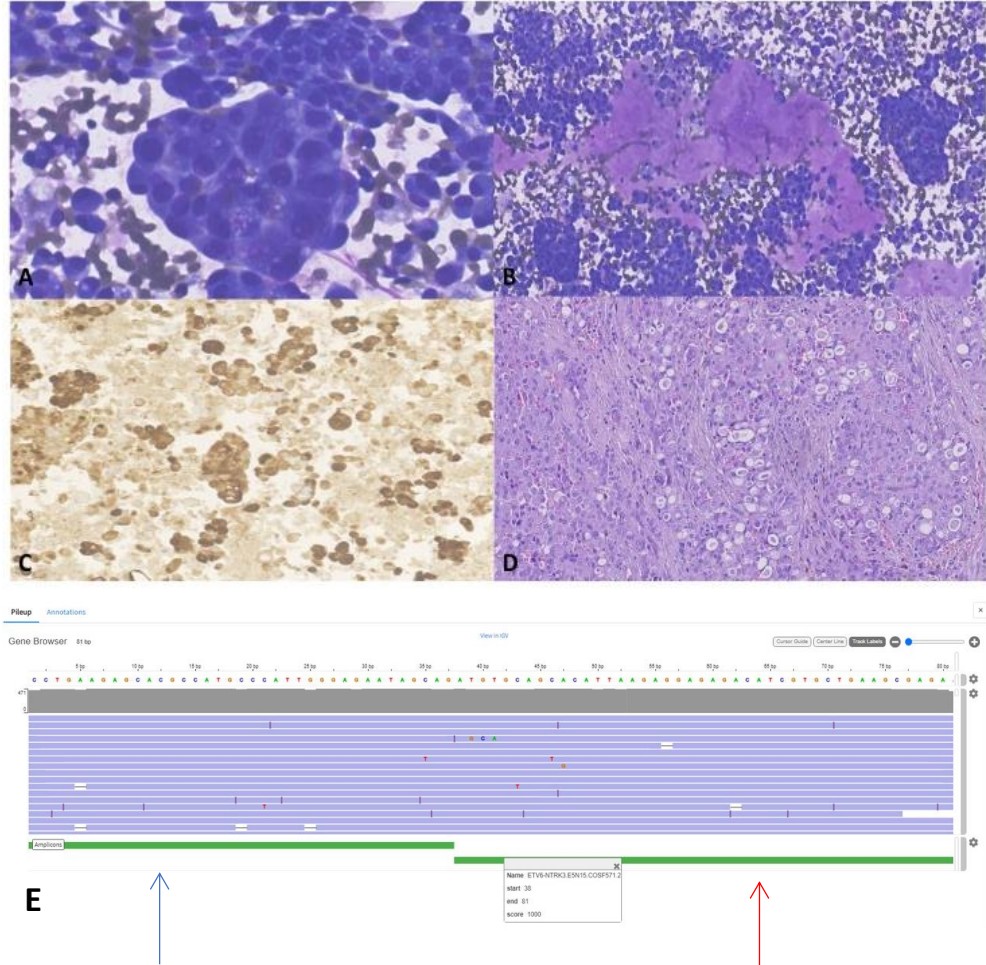

**Figure 3.** (**A**) Solid and papillary groups of polygonal cells with a low N:C ratio, vacuolated cytoplasm and indistinct cell borders, DQ, 20×; (**B**) scattered metachromatic dense extracellular material, DQ, 10×; (**C**) diffuse nuclear and cytoplasmic positivity for anti-S100 immunostaining, 10×; (**D**) histologic sample of a neoplasm with a solid and microcystic pattern comprising polygonal cells with eosinophilic or vacuolated cytoplasm and nuclei with mild pleomorphism, H/E, 20×; (**E**) identification by next generation sequencing of ETV6–NTRK3 gene fusion as a result of fusion of the ETV6 gene (blue arrow) on chromosome 12 with the NTRK3 gene (red arrow) on chromosome 15, translocation visualized through the software gene browser provided by Thermo Fisher Scientific Genexus.

Unfortunately, the patient was affected by frequent local relapses. An additional FNA was requested for the last relapsed mass, showing cytomorphological features similar to that observed on the primary lesion. Thus, to assess the eligibility of this patient for a targeted therapeutic regimen, the neoplastic cells were scraped from the FNA smear to allow RNA extraction and NGS testing through use of an Oncomine Precision Assay (OPA, Thermo Fisher Scientific). The minimum quantity of RNA input needed is 10 ng. In our clinical practice, we use this panel on several types of cytological specimens, both direct smears and cell blocks (CBs). An *ETV6–NTRK3* translocation was detected, which is the most frequent genomic alteration harbored by secretory carcinomas. The patient was then treated with larotrectinib, a kinase inhibitor that has demonstrated efficacy in TRK fusion-positive salivary gland neoplasms with a favorable safety profile [11]. To date, the patient is still under treatment with larotrectinib, showing an excellent performance status with no evidence of further relapse. This case reveals the usefulness of molecular testing performed on FNA in patients affected by advanced salivary gland neoplasms, avoiding more invasive procedures and allowing at the same time proper therapy.

## 5. Case 4

Here we present a case in which molecular tests helped with the management of a patient affected by thyroid neoplastic disease. A 35-year-old woman was referred to our outpatient FNA clinic after a thyroid ultrasound scan that detected a 14 mm thyroid nodule. The smears revealed a colloidal hematic background with numerous histiocytes and scattered groups of thyrocytes, with complex architectural and cytological atypia, including papillary-shaped group and atypical "histiocytoid" cells; these latter are often associated with papillary thyroid carcinoma with cystic degeneration (Figure 4A). Thus, the FNA was diagnosed as atypia of undetermined significance with nuclear atypia (AUS-nuclear) according to the new edition of the Bethesda System for Reporting Thyroid Cytopathology (TBSRTC) [12]. In our clinical cytology practice, after an indeterminate diagnosis, the leftover material in the needle hub is rinsed in a nuclease-free water filled vial to carry out the 7-gene test as previously described [13]. Briefly, the procedure was performed by the Real Time QuantStudio 5 platform (Applied Biosystems; Thermo Fisher) using the EntroGen Thyroid Cancer Mutation Analysis Panel Kit (EntroGen, Inc., Woodland Hills, CA, USA), which is able to detect *BRAF* V600E; *KRAS* codon 12 and 13; *NRAS* codon 61; and *HRAS* codon 12, 13, and 61 point mutations. This assay requires at least a DNA content of 1 ng for each sample.

Concerning this, in our diagnostic routine, we perform this analysis on cytological (fine needle aspirated material collected in a plastic tube or fixed inside ThinPrep®, Hologic) and fixed formalin and paraffin embedded specimen (FFPE samples) with a cellullarity content of at least 5–10%. A *BRAF* p.V600E mutation in exon 15 was detected (Figure 4B). *BRAF* p.V600E mutation is associated with a >95% probability of papillary carcinoma and surgical referral is usually advised [12]. Thus, this patient was referred for a total oncologic thyroidectomy; the histopathological examination confirmed the occurrence of a papillary thyroid carcinoma.

Molecular testing in thyroid cytopathology allows either more accurate pre-operative risk stratification for patients with indeterminate FNA cytology [12,15] or the detection of targetable molecular alterations in patients affected by advanced, radioactive iodine refractory carcinomas [16]. In particular, the identification of clinically relevant alterations implicated in thyroid oncogenesis, such as the *BRAF* p.V600E mutation, is essential to help differentiate benign from malignant thyroid nodules in cases of indeterminate cytology, and to allow targeted therapy in patients with advanced radioiodine refractory thyroid cancer.

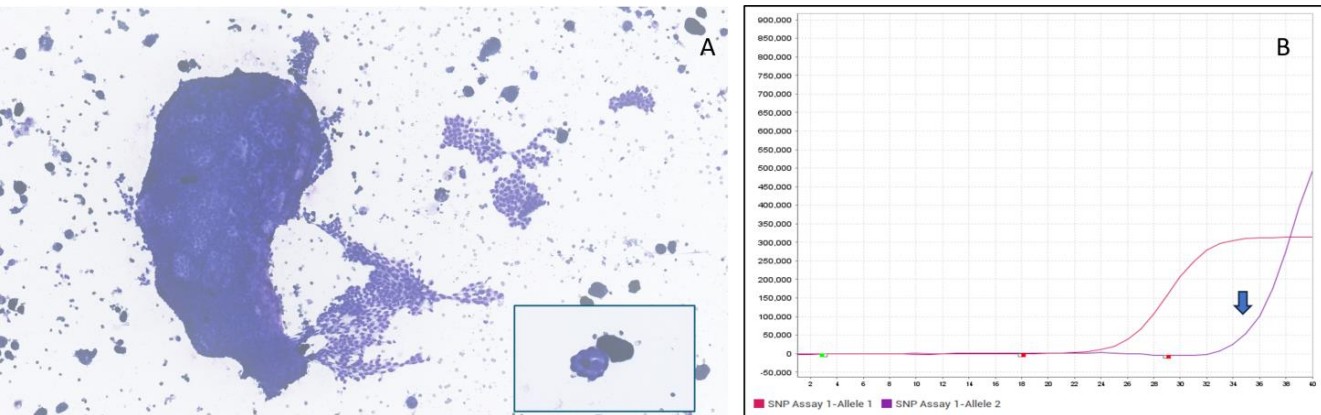

**Figure 4.** (**A**) Scattered groups of thyrocytes, with complex architectural and cytological atypia, including papillary-shaped group and atypical "histiocytoid" cells (inset) (Papanicolaou staining, ×20 magnification). (**B**) Representative amplification curves by real-time polymerase chain reaction (PCR) for DNA samples from wild-type BRAF (red curve) and *BRAF* V600E single-nucleotide mutation specimens (purple curve). Compared to wild-type *BRAF*, the *BRAF* V600E mutation sample required a greater number of amplification cycles (x axis) to reach a maximal signal intensity. The second case concerned a 67-year-old man who came to our institution for a routine check-up following a total thyroidectomy for a radioiodine refractory high-grade differentiated papillary thyroid carcinoma (HGDTC). The CT scan showed a neck mass and a lung metastasis (Figure 5A,B). A FNA was performed on the neck mass, confirming the recurrence of the HGDTC. RT-PCR analysis performed on the FNA detected a *BRAF* p.V600E; thus, the patient was deemed eligible for combination therapy with BRAF and MEK inhibitors [14]. The follow-up CT scan after the first three months of therapy revealed a complete regression of the neck lump and a volumetric reduction of the lung metastasis (Figure 5C,D).

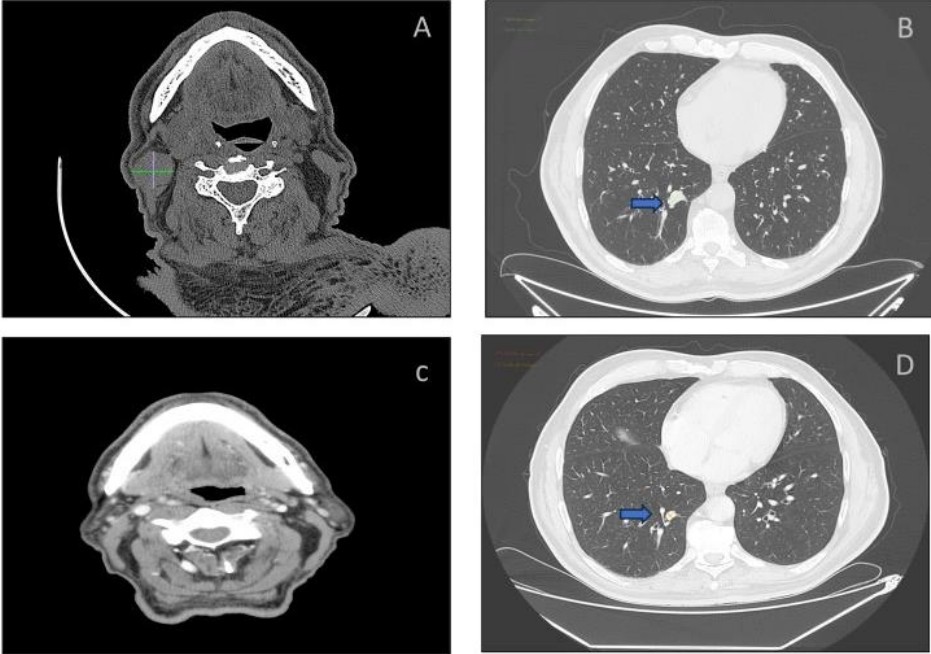

**Figure 5.** CT scans confirming the presence of a neck mass (**A**) and a metastatic nodule in the lung ((**B**), arrow). CT scans after three months of BRAF and MEK inhibitors treatment showing complete lump regression in the neck (**C**) and reduction of lung metastasis volume in the lung (**D**), arrow.

### 6. Case 5

As shown for the Case 1 we received, as a referral center for predictive testing, six air-dried smears and a CB, obtained from a CT-guided FNA on a 47 mm lung nodule in a 66-year-old man. No other clinical information was available and clinical suspicion of a primitive lung neoplasm was reported, prompting the request for the molecular characterization of clinically relevant biomarkers. The Diff-Quik stained smears showed necrosis and numerous atypical epithelial cells featuring prominent nucleoli arranged in solid and trabecular groups; chromatin "crushing" was also observed. The CB sections confirmed high cellularity with solid and trabecular groups of atypical epithelial cells (Figure 6A,B). The first panel of immunostaining showed negativity for thyroid transcription factor 1 (TTF-1), p40 and chromogranin. Thus, a second panel of immunostaining was performed to refine the diagnosis. Unfortunately, synaptophysin, CD56, cytokeratin 7 (CK7) and CK20 were also negative. Based on microscopic and ICC features, a diagnosis of non-small cell lung carcinoma NOS was hypothesized. It has been widely shown that this variant is strongly associated with metastatic prostate cancer [17] and resistance to antiandrogen drugs such as enzalutamide and abiraterone [18]. Therefore, an additional ICC analysis was ordered, revealing diffuse positivity for prostate specific antigen (PSA), confirming the metastatic origin of the lung lesion (Figure 6C). After further inquiring, the oncologist confirmed a history of prostatic adenocarcinoma going back to ten years before the development of the lung lesion.

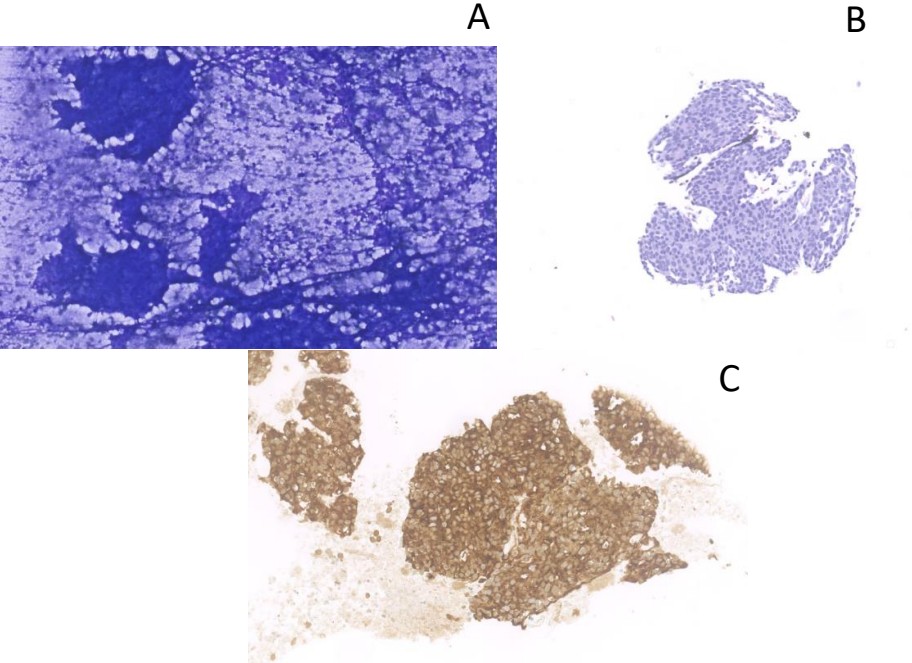

**Figure 6.** Diff-Quik stained slide (**A**) and cell block section (**B**) showing solid and trabecular groups of atypical epithelial cells. (**C**) Prostate specific antigen (PSA) immunostaining showed intense and diffuse positivity.

In conclusion, this case highlighted how an integrated morphological, immunophenotypical, and molecular approach helped the cytopathologist in a challenging case, and strongly support the necessity of a close collaboration among all of the healthcare specialists (cytopathologist, oncologist, and molecular biologist) to ensure the best management of cancer patients and avoid unnecessary delays in the diagnostic process.

### 7. Conclusions

These cases highlight the challenges of modern molecular cytopathology. Each case demonstrated how the integration of morphology and molecular data can make a major

difference in the clinical management and the treatment of patients. In particular, several practical lessons can be learned:

- Although the preservation of sufficient tissue for biomarker testing is crucial, a limited immunocytochemical panel is advised to properly classify poorly differentiated non-small cell lung cancer. Moreover, a comprehensive approach involving simultaneous NGS and ICC can optimize time and the use of scant material;
- ROSE significantly enhances FNA diagnosis, streamlining the process for ancillary techniques, which are especially crucial in challenging cases; the cytological material can be exploited for different kinds of testing. In fact, both NGS and FISH can highlight the presence of pathognomonic gene fusions in cytological material, avoiding the need for more invasive procedures;
- Molecular testing in thyroid cytopathology enables precise pre-operative risk assessment for patients with indeterminate fine-needle aspirate (FNA) cytology or the identification of targetable molecular alterations in those with advanced thyroid carcinomas.

However, these advanced molecular techniques applied to cytopathological specimens should be always be supported by close interdisciplinary collaboration.

**Author Contributions:** Conceptualization, E.V., C.B., M.D.L., S.R.-C. and G.T.; Methodology, all authors; Software, all authors; Validation, all authors; Formal Analysis, all authors; Investigation, all authors; Resources, all authors; Data Curation, all authors; Writing—Original Draft Preparation, all authors; Writing—Review & Editing, all authors; Visualization, all authors; Supervision, E.V., C.B., M.D.L., S.R.-C. and G.T.; Project Administration, E.V., C.B., M.D.L., S.R.-C. and G.T. All authors have read and agreed to the published version of the manuscript.

**Funding:** No funding or sponsorship was received for the publication of this article.

**Institutional Review Board Statement:** Not applicable.

**Informed Consent Statement:** Written informed consent has been obtained from the patient(s) to publish this paper.

**Data Availability Statement:** The data presented in this study are available on request from the corresponding author.

**Conflicts of Interest:** The authors declare no conflicts of interest.

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
