# Peer review of "How Molecular and Ancillary Tests Can Help in Challenging Cytopathology Cases: Insights from the International Molecular Cytopathology Meeting"

_jmp, doi:10.3390/jmp5020015_

Round 1
Reviewer 1 Report
Comments and Suggestions for Authors
The images are difficult for the non-specialist, fuzzy and sometimes overexposed. Annotations would help clarify.
Author Response
The images are difficult for the non-specialist, fuzzy and sometimes overexposed. Annotations would help clarify.
DOne.
Reviewer 2 Report
Comments and Suggestions for Authors
The article is well written, easy to understand and follow.
There are minor things to correct:
-line 103, 104 “a final diagnosis of metastasis from a squamous cell carcinoma of pulmonary origin was performed“ - “was performed” there must be an error in this sentence and choice of words so please correct it.
Figure 2 D there is disproportion in the left and right side in this image, so the right side are magnified figures of two nuclei which have been inserted to show translocation-clarify and insert arrows to show signals that indicate translocation.
Figure 4 A where is the insert that has been indicated in the description? 4 B the resolution of image showing PCR amplification is too low, numbers on x and y axis are not visible, and describe in more detail what each line in PCR image represents.
Figure 6C very poor resolution of NGS analysis data, nothing can be seen in this image because it is blurry and try to be more precise and indicate in arrow in some other way which part of image shows the intragenic fusion visible.
I recommend it for publication after these minor changes.
Author Response
The article is well written, easy to understand and follow.
There are minor things to correct:
-line 103, 104 “a final diagnosis of metastasis from a squamous cell carcinoma of pulmonary origin was performed“ - “was performed” there must be an error in this sentence and choice of words so please correct it.
Done.
Figure 2 D there is disproportion in the left and right side in this image, so the right side are magnified figures of two nuclei which have been inserted to show translocation-clarify and insert arrows to show signals that indicate translocation.
We Thank the Reviewer and we modify Figure 2 in the revised version of the paper.
Figure 4 A where is the insert that has been indicated in the description? 4 B the resolution of image showing PCR amplification is too low, numbers on x and y axis are not visible, and describe in more detail what each line in PCR image represents.
We Thank the Reviewer and we modify Figure 4 legend in the revised version of the paper.
Figure 6C very poor resolution of NGS analysis data, nothing can be seen in this image because it is blurry and try to be more precise and indicate in arrow in some other way which part of image shows the intragenic fusion visible.
We Thank the Reviewer and we modify Figure 6 in the revised version of the paper.
I recommend it for publication after these minor changes.
Reviewer 3 Report
Comments and Suggestions for Authors
To serve as a source of insight to pathologists who would learn to integrate the molecular pathology and morphological evaluation of cytopathology specimens, the authors present 5-6 illustrative cytopathology cases that were discussed during the 2023 Annual International Molecular Cytopathology Meeting. Although the cases are interesting, none of them discuss the principles and practical aspects of performing and interpreting molecular pathology analyses of valuable cytopathology specimens in sufficient detail such that a pathologist would know and understand the basics of molecular pathology testing in those cases.
Cases 1-5: for each heading, mention the name of the tissue or diagnosis to guide the reader.
Fig. 1-5; most of the images are blurry. Please provide higher resolution images.
Case 1: Fig. 1 does not illustrate the data output of the Oncomine Precision Assay. What tumor content and what minimum quantity of DNA/RNA are needed for the test to be performed? Can the test be performed on FFPE and liquid biopsies? Regarding the immunocytochemistry, mention that immunoperoxidase detection with diaminobenzidine was used. Define tumor proportion score.
Case 2: Line 11-112 and Fig. 2D, explain the principle of the HEY1-NOCA2 translocation FISH probe and how to interpret the significance of the blue, red and yellow intracellular signals. Fig. 2A-D, indicate the magnification.
Case 3: Line 156-157, what tumor content and what minimum quantity of RNA are needed to perform the NGS Archer FusionPlex Lung? What kind of cytopathology specimen can be submitted for this kind of analysis: cytology smear, fine needle aspiration? Describe the data output showing the ETV-NTRK3 translocation.
Case 4A and B: Line 185-186 and Fig. 4B, describe what detection method and probe were used for the RT-PCR, and how to read the result of the RT-PCR amplification. What tumor content and what minimum quantity of DNA are needed for this test? What type of cytopathology specimen can be submitted for this kind of analysis?
Case 5: Line 233-234 and Fig. 6C, describe how to read the results of the androgen receptor splice variant by next generation sequencing of RNA specimen. Again, what tumor content, minimum quantity of RNA and type of cytopathology specimen are required for this kind of analysis?
References: please ensure that the references are complete and follow a uniform citation format.
Author Response
To serve as a source of insight to pathologists who would learn to integrate the molecular pathology and morphological evaluation of cytopathology specimens, the authors present 5-6 illustrative cytopathology cases that were discussed during the 2023 Annual International Molecular Cytopathology Meeting. Although the cases are interesting, none of them discuss the principles and practical aspects of performing and interpreting molecular pathology analyses of valuable cytopathology specimens in sufficient detail such that a pathologist would know and understand the basics of molecular pathology testing in those cases.
Cases 1-5: for each heading, mention the name of the tissue or diagnosis to guide the reader.
Fig. 1-5; most of the images are blurry. Please provide higher resolution images.
Case 1: Fig. 1 does not illustrate the data output of the Oncomine Precision Assay. What tumor content and what minimum quantity of DNA/RNA are needed for the test to be performed? Can the test be performed on FFPE and liquid biopsies? Regarding the immunocytochemistry, mention that immunoperoxidase detection with diaminobenzidine was used. Define tumor proportion score.
We Thank the Reviewer and we modify accordingly the revised version of the paper.
Case 2: Line 11-112 and Fig. 2D, explain the principle of the HEY1-NOCA2 translocation FISH probe and how to interpret the significance of the blue, red and yellow intracellular signals. Fig. 2A-D, indicate the magnification.
We Thank the Reviewer and we modify Figure 2 and the text in the revised version of the paper.
Case 3: Line 156-157, what tumor content and what minimum quantity of RNA are needed to perform the NGS Archer FusionPlex Lung? What kind of cytopathology specimen can be submitted for this kind of analysis: cytology smear, fine needle aspiration? Describe the data output showing the ETV-NTRK3 translocation.
We Thank the Reviewer and we modify accordingly the revised version of the paper.
In particular, the NGS Archer FusionPlex Lung was reported for a typing error.
We used the Oncomine Precision Assay (OPA, Thermo Fisher scietific) gene panel that is able to detect several alterations(mutations, fusions and Copy Number Variations) in 50 genes, among which NTRK3. The minimum quantity of RNA input needed is 10ng. In our clinical practice, we use this panel on several cytological speciments both direct smears and cell-blocks (CBs).
Case 4A and B: Line 185-186 and Fig. 4B, describe what detection method and probe were used for the RT-PCR, and how to read the result of the RT-PCR amplification. What tumor content and what minimum quantity of DNA are needed for this test? What type of cytopathology specimen can be submitted for this kind of analysis?
We Thank the Reviewer and we modify accordingly the revised version of the paper.
Case 5: Line 233-234 and Fig. 6C, describe how to read the results of the androgen receptor splice variant by next generation sequencing of RNA specimen. Again, what tumor content, minimum quantity of RNA and type of cytopathology specimen are required for this kind of analysis?
We Thank the Reviewer and we modify accordingly the revised version of the paper.
References: please ensure that the references are complete and follow a uniform citation format.
We Thank the Reviewer and we modify accordingly the revised version of the paper.
Round 2
Reviewer 3 Report
Comments and Suggestions for Authors
Thank you for making improvements to the manuscript. Nonetheless, additional revisions and corrections are required, especially with regard to explaining the methodology and the interpretation of the molecular pathology tests.
Line 3: change “helped” to “can help”.
Line 39, 42, 45, 49-51: please avoid text repetition.
Fig. 1 legend: what do you mean by “solid group”?
Fig. 2: duplicate images. Fig. 2D legend: regarding the “HEY1-NOCA2” probe, do you mean “using a HEY1/NCOA2 translocation FISH probe to detect HEY1-NCOA2 fusion. The fused signal (visible as one yellow dot (red arrow), instead of green dot/red dot side by side for the separate probes) is seen in positive cells."?
Fig. 3: duplicate images. Fig. 3D legend: regarding the “ETV6(5) - NTRK3(15) translocation”, do you mean “identification by next generation sequencing of ETV6-NTRK3 gene fusion as a result of fusion of the ETV6 gene on chromosome 12 with the NTRK3 gene on chromosome 15.” Please label the sequences corresponding to 5’ region of ETV6 and 3’ region of NTRK3 on the interrupted green line.
Fig. 4A: clearly delineate the inset. Fig. 4B: to guide to readers with result interpretation, consider writing “Representative amplification curves by real-time polymerase chain reaction (PCR) for DNA samples from wild-type BRAF (red curve) and BRAF V600E single-nucleotide mutation specimens (purple curve). Compared to wild-type BRAF, BRAF V600E mutation sample requires a greater number of amplification cycles (x axis) to reach a maximal signal intensity.”
Fig. 5: how long (months, years?) after what doses of BRAF and MEK inhibitors were panels C and D obtained?
Fig. 6: duplicate images. Fig. 6C legend: actually androgen receptor splicing variant AR-V7 originates from contiguous splicing of AR exons 1, 2, and 3 with the cryptic exon 3 (CE3) present within the canonical intron 3 of the AR gene. Not sure what is meant by “AR(3):AR(4)”. Please label the sequences corresponding to AR exons 2, 2 and 3 and CE3 on the interrupted green line.
Author Response
Thank you for making improvements to the manuscript. Nonetheless, additional revisions and corrections are required, especially with regard to explaining the methodology and the interpretation of the molecular pathology tests.
We thank the reviewer for the suggestions; we modify the present version of the manuscript accordingly.
Line 3: change “helped” to “can help”.
Done
Line 39, 42, 45, 49-51: please avoid text repetition.
We have reshaped the sentence.
Fig. 1 legend: what do you mean by “solid group”?
We mean “cohesive group”; we modify the legend accordingly.
Fig. 2: duplicate images. Fig. 2D legend: regarding the “HEY1-NOCA2” probe, do you mean “using a HEY1/NCOA2 translocation FISH probe to detect HEY1-NCOA2 fusion. The fused signal (visible as one yellow dot (red arrow), instead of green dot/red dot side by side for the separate probes) is seen in positive cells."?
Done
Fig. 3: duplicate images. Fig. 3D legend: regarding the “ETV6(5) - NTRK3(15) translocation”, do you mean “identification by next generation sequencing of ETV6-NTRK3 gene fusion as a result of fusion of the ETV6 gene on chromosome 12 with the NTRK3 gene on chromosome 15.” Please label the sequences corresponding to 5’ region of ETV6 and 3’ region of NTRK3 on the interrupted green line.
Done
Fig. 4A: clearly delineate the inset. Fig. 4B: to guide to readers with result interpretation, consider writing “Representative amplification curves by real-time polymerase chain reaction (PCR) for DNA samples from wild-type BRAF (red curve) and BRAF V600E single-nucleotide mutation specimens (purple curve). Compared to wild-type BRAF, BRAF V600E mutation sample requires a greater number of amplification cycles (x axis) to reach a maximal signal intensity.”
Done
Fig. 5: how long (months, years?) after what doses of BRAF and MEK inhibitors were panels C and D obtained?
We added the duration of therapy to the legend (3 months)
Fig. 6: duplicate images. Fig. 6C legend: actually androgen receptor splicing variant AR-V7 originates from contiguous splicing of AR exons 1, 2, and 3 with the cryptic exon 3 (CE3) present within the canonical intron 3 of the AR gene. Not sure what is meant by “AR(3):AR(4)”. Please label the sequences corresponding to AR exons 2, 2 and 3 and CE3 on the interrupted green line.
Done